# Adapt in Contexts: Retrieval-Augmented Domain Adaptation via In-Context Learning

**Quanyu Long**[1]  **Wenya Wang**[1]  **Sinno Jialin Pan**[1,2]

[1]Nanyang Technological University, Singapore
[2]The Chinese University of Hong Kong

{quanyu001, wangwy}@ntu.edu.sg    sinnopan@cuhk.edu.hk

## Abstract

Large language models (LLMs) have showcased their capability with few-shot inference known as in-context learning. However, in-domain demonstrations are not always readily available in real scenarios, leading to cross-domain in-context learning. Besides, LLMs are still facing challenges in long-tail knowledge in unseen and unfamiliar domains. The above limitations demonstrate the necessity of Unsupervised Domain Adaptation (UDA). In this paper, we study the UDA problem under an in-context learning setting to adapt language models from the source domain to the target domain without any target labels. The core idea is to retrieve a subset of cross-domain elements that are the most similar to the query, and elicit language model to adapt in an in-context manner by learning both target domain distribution and the discriminative task signal simultaneously with the augmented cross-domain in-context examples. We devise different prompting and training strategies, accounting for different LM architectures to learn the target distribution via language modeling. With extensive experiments on Sentiment Analysis (SA) and Named Entity Recognition (NER) tasks, we thoroughly study the effectiveness of ICL for domain transfer and demonstrate significant improvements over baseline models.

## 1  Introduction

Large Language Models (LLMs) have demonstrated remarkable success in various tasks via in-context learning (ICL) with task instructions and few-shot demonstrations (input-label pairs) (Zhao et al., 2021; Liu et al., 2022; Min et al., 2022), eliminating the need for fine-tuning from task-specific labels. Nevertheless, in-domain demonstrations are usually absent in real scenarios since the target domain labels are unavailable. Sourcing labeled examples from other domains may suffer from huge syntactic and semantic domain shifts. Moreover,

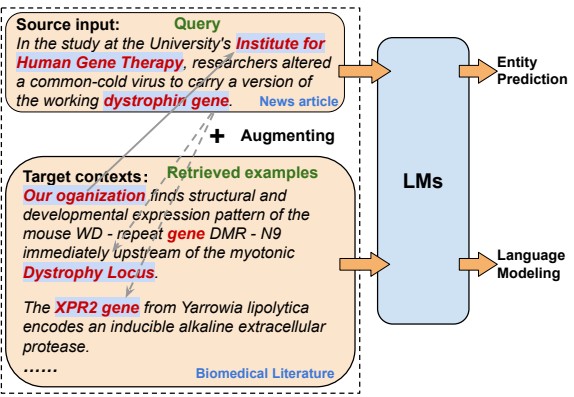

Figure 1: A motivating example of retrieval-augmented in-context adaptation for NER: biomedical texts retrieved from the target unlabeled domain will serve as demonstrative contexts to help LMs correctly predict entities "Institute for Human Gene Therapy" and "dystrophin gene" (solid arrow). The language model transfers the knowledge to the target domain to identify unknown entities with a similar structure like "XPR2 gene" or "dystrophy locus" by learning target distribution with language modeling (dotted arrow).

LLMs are prone to generating unpredictable outputs in undesired formats, and they are struggling with long-tail knowledge for unseen and unfamiliar domains where topics and genres are less frequently encountered in the training corpus (Asai et al., 2023). Therefore, the limitations above call for effective adaptation strategies to transfer knowledge of LMs from a labeled source domain to the unlabeled target domain, known as Unsupervised Domain Adaptation (UDA).

To bridge the domain gap, UDA aims to adapt models that learn domain-agnostic features from labeled source samples and unlabeled target samples. Some studies have proposed discrepancy measures to align source and target distributions in the representation space (Ganin et al., 2016; Ye et al., 2020; Long et al., 2022). However, these methods mainly focus on feature alignment and only apply to encoder-based LMs. Other studies focus on adaptive pre-training including an additional post

pre-training phase of masked language modeling (MLM) on target unlabeled data to learn the target domain distribution (Han and Eisenstein, 2019; Karouzos et al., 2021). However, different training phases make the learned diverse knowledge hard to remember, and such methods are also only applicable to encoder-only LMs which are usually smaller in scale. Therefore, few studies have investigated how to update knowledge of unfamiliar domains for larger LMs (e.g., decoder-only LMs). And few studies try to relate source-labeled samples to target unlabeled examples in a single training stage, while vast amounts of target unlabeled data can serve as a knowledge-rich datastore.

In this paper, we propose to retrieve similar examples from the target unlabeled corpus to serve as the context of a source query and perform adaptive in-context learning by concatenating the source query and target contexts as the input prompt. The core idea is to elicit LMs to learn target distribution and discriminative task signals simultaneously with the retrieved cross-domain examples. Fig. 1 shows an illustrative example. For each input from the source domain, we compose its context with semantically similar texts retrieved from the target unlabeled domain to enrich semantics and reduce the domain difference in the surface form. Then the model will learn the task discrimination taking both the source input and the target context. To further mitigate domain shift, we propose to learn the target distribution using the language modeling mechanism (causal or masked language modeling) simultaneously by predicting tokens from the target context, which acts as a proxy to the target distribution. Combining the two goals encourages the model to learn domain-agnostic and task-aware information which is beneficial for knowledge transfer.

We propose a domain-adaptive in-context learning (DAICL) framework for different LM architectures, including encoder-only and decoder-only models, and observe consistent advantages. To account for the architectural difference, we devise distinct prompting and fine-tuning strategies. For the encoder-only model, we append contexts retrieved from the target domain to each source input. The model is trained to predict source input labels and masked tokens in the appended contexts. For the decoder-only model, we instead prepend examples before the source input. The model is trained to predict each token autoregressively in the prompt

as well as the response output.

Overall, we make the following contributions:

- We propose domain-adaptive in-context learning with retrieval augmentation in which we mix the source input and semantically rich target contexts to learn two in-context objectives simultaneously;

- We proposed a unified framework with efficient prompting and fine-tuning strategies accounting for different architectures (encoder-only LMs and decoder-only LMs);

- We thoroughly study the effectiveness of in-context learning for UDA. Our experiments surprisingly reveal that retrieving out-of-distribution demonstrations fails for LLMs' few-shot inference and fine-tuning is still beneficial for domain adaptation.

## 2 Problem Definition

Consider a scenario where we have access to two distinct domains: a source domain and a target domain. The source domain dataset, denoted as $\mathcal{D}^S$, consists of $n$ labeled data sampled i.i.d. from the source distribution, $\mathcal{D}^S = \{x_i^S, y_i\}_{1,...,n}$, where $x_i^S$ represents sequences of tokens, $y_i$ represents the corresponding label. On the other hand, the unlabeled target domain dataset, denoted as $\mathcal{D}^T = \{x_j^T\}_{1,...,m}$, comprises $m$ unlabeled data points, which are also sampled i.i.d. from the target domain. The primary objective of Unsupervised Domain Adaptation (UDA) is to adapt the knowledge learned from the source domain in such a way that allows it to generalize on the target domain effectively. This adaptation process involves leveraging the unlabeled data from the target domain to learn the target distribution and mitigate the domain shift.

This paper focuses on the UDA problem over two application scenarios: Named Entity Recognition (NER)[1] and Sentiment Analysis (SA). We describe our method and pivot discussions around these two tasks in the following sections.

## 3 Method

We propose a novel framework, Domain Adaptive In-Context Learning (DAICL), capable of training

---

[1]In our work, we only need to predict the entity spans, ignoring the entity type, because the label spaces for different domains are different when considering entity type. However, we still refer to this task as NER for short.

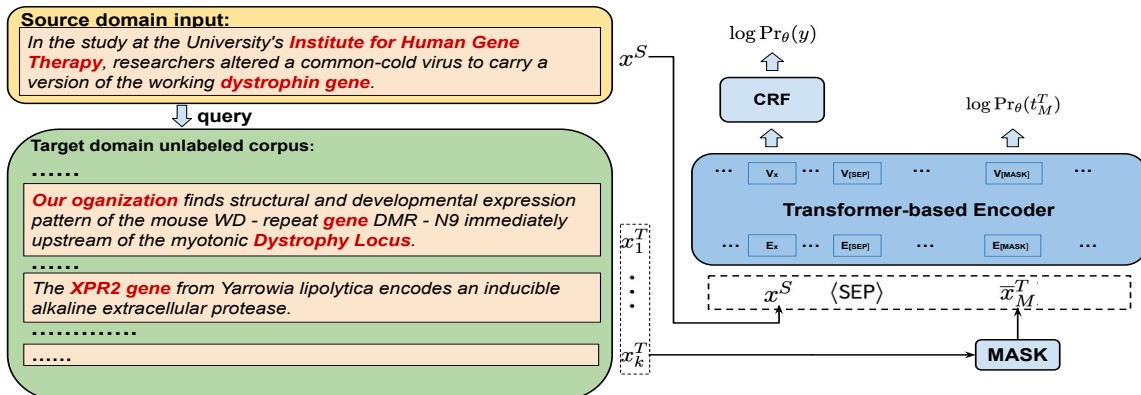

Figure 2: An overview of training encoder-only NER models with retrieved context via in-context learning.

LMs to adapt with the help of retrieved contexts. We begin by introducing the overall framework in Section 3.1. Next, we present specific designs for Encoder-only language models in Section 3.2; and Decoder-only language models in Section 3.3. For decoder-only models, we present two settings: inference-only (Section 3.3.1) and fine-tuning (Section) 3.3.2.

## 3.1 In-Context Adaptation

The term *In-Context Learning* has been commonly referred to as few-shot prompting in LLMs. To be clear, in this work, we instead use *In-Context Learning* to emphasize the idea of learning a model with semantically rich contexts. Here *context* should be differentiated with *demonstration*, the latter one represents input-label pairs in few-shot prompting. Under the setting of UDA where target labels are not accessible, *context* is composed of input-only examples from the unlabeled target domain. Next, we present an overall framework to construct suitable contexts and adapt LMs with suitable contexts.

### 3.1.1 Context Construction with Retrieval

Given an input sentence from the source domain, we first search for semantically similar examples from the unlabeled target domain. This is analogous to retrieval and re-rank given a search query. Retrieval-augmented LM approaches (Guu et al., 2020; Lewis et al., 2020; Asai et al., 2023) apply a parametrized dense retriever to train with the task model. In this paper, we fix the retriever part and use the off-the-shelf scoring language models. For the SA task, we use SimCSE (Gao et al., 2021) which produces semantically meaningful sentence embeddings after being trained with contrastive learning (Chen et al., 2020; He et al., 2020). Here cosine similarity is used to retrieve

top-ranked (most similar) examples from the target domain. For NER, we use BERTScore (Zhang et al., 2020; Wang et al., 2021), because it gives a metric for each sentence based on the similarity of token representation, which is more crucial for the task of NER.

Specifically, given a source sentence $x^S$ paired with label $y$, we retrieve top-$k$ relevant chunks of texts from the target unlabeled dataset $\mathcal{D}^T$. The retrieved examples are denoted as $\overline{x}^T = \{x_1^T, \cdots, x_k^T\}$ which will serve as the contexts to enrich the semantics for the source input.

### 3.1.2 Domain-Adaptive In-Context Learning

With the retrieved context consisting of $k$ most semantically similar examples to the source input, we seek a strategy to integrate this context into the source input and design a training objective that could learn target distribution and at the same time be able to discriminate the task label. To this end, we propose to combine the following two objectives given the concatenated text sequences $[x^S; \overline{x}^T]$. Objective 1: **In-context Task Learning** – a supervised task to predict the task label $y$. Objective 2: **In-context Language Modeling** – a token prediction task to predict tokens from the target context $\overline{x}^T$:

$$\mathcal{L}_{Sup}(\theta) = -\log \Pr_\theta \left( y \big| x^S, \overline{x}^T \right); \quad (1)$$
$$\mathcal{L}_{LM}(\theta) = -\log \Pr_\theta \left( t_i^T \big| x^S, \overline{x}^T \right), t_i^T \in \overline{x}^T, \quad (2)$$

where $\theta$ represents the parameters for a language model. Ideally, the first objective (1) aims to learn task discrimination with the help of context. Note that unlike single-domain task prediction which only takes $x^S$ as input, here we augment the source input with target contexts to learn task-aware information across domains. The second objective (2)

encourages the model to learn the target distribution by predicting tokens in the target context $\overline{x}^T$. By mixing with a source input, the model learns to fuse the distributions from two different domains in order to bridge the domain gap. When combining these two objectives, we expect that the model learns task-aware knowledge that is indistinguishable from the two domains.

## 3.2 Encoder-only LMs with ICL

This section describes domain-adaptive in-context learning with encoder-only LMs, e.g., BERT (Devlin et al., 2019). As discussed in Section 3.1, for each input $x^S$, we first retrieve top-$k$ sentences $\overline{x}^T$ from the target domain as the context for $x^S$. For encoder-only models, the retrieved sentences are then concatenated at the end of the source input.

$$[x^S; \overline{x}^T] = \left[ x^S; \langle \mathsf{SEP} \rangle ; x_1^T; \cdots ; x_k^T \right], \quad (3)$$

where $\langle \mathsf{SEP} \rangle$ is a separation token.

To perform in-context learning, recall from Section 3.1, two objectives (language modeling and task learning) are involved. An overview of the training process for encoder-only models on the NER task is shown in Fig. 2. For the language modeling objective, we perform unsupervised Masked Language Modeling (MLM) on the target domain. We randomly sample $15\%$ tokens from the target context $[x_1^T; \cdots ; x_k^T]$ and replace them with the $[\mathsf{MASK}]$ token. We denote the set of indices for the masked tokens as $M$ and the original ground-truth tokens for these masked positions are referred to as $t_M^T = \{t_i | i \in M\}$. The masked input becomes $[x; \overline{x}_M^T]$, where $\overline{x}_M^T$ denotes the collection of target contexts after masking. With the bidirectional structure of the encoder-only LMs, the representation for each masked token in the target domain encodes both the target context and the source input. As such, the MLM objective encourages the encoder to learn the target distribution that is indistinguishable from the source domain.

For the task objective, we use different prediction mechanisms for different tasks. For SA, we use average pooling on top of each token in the source input $x^S$ before being fed into the classifier. For NER, we apply an additional CRF layer (Ma and Hovy, 2016; Lample et al., 2016) on top of the LM feature which is a common practice for token-level classifications.

Formally, the joint objective is to minimize the negative log-likelihood of the ground truth task label $y$ and masked tokens $t_M^T$:

$$\min_\theta \sum_{(x^S, y) \sim \mathcal{D}^S} - \big[ \log \mathrm{Pr}_\theta(y | x^S, \overline{x}_M^T) + \lambda \log \mathrm{Pr}_\theta(t_M^T | x^S, \overline{x}_M^T) \big], \quad (4)$$

where $\lambda$ represents a scaling factor.

## 3.3 Decoder-only LMs with ICL

Recently, decoder-only LMs have received excessive attention and have motivated continuous developments to scale up in order to solve various NLP tasks under zero-shot or few-shot settings, such as GPT-3 (Brown et al., 2020), LLaMA (Touvron et al., 2023), and ChatGPT. Despite the increasing scalability, they are still prone to producing unpredictable outputs in undesired formats. For example, ChatGPT gives subpar performance for NER (see Table 1). This reflects the necessity of decoder-only LMs for learning to adapt to the target domain.

### 3.3.1 Cross-Domain Few-Shot Inference

Recent works show that providing few-shot ICL demonstrations (input-label pairs) contributes to performance gains (Zhao et al., 2021; Liu et al., 2022; Min et al., 2022). However, there are no in-distribution demonstrations available when performing inference on the unlabeled target domain. Therefore, in many real scenarios, we often select out-of-distribution (OOD) input-label pairs from another domain irrespective of the possible huge domain shift from the target query. In UDA, we have access to the entire labeled source dataset, thus we could retrieve similar demonstrations from the source domain given a target query. We provide prompts and examples in Fig. 3 showing how to use retrieved input-label pairs from the source domain as demonstrations.

In our experiments with ChatGPT (see Table. 1 and Table. 2), surprisingly we find that retrieving OOD demonstrations fails in most adaptation scenarios; even randomly sampling cross-domain demonstrations can bring non-trivial performance gains comparing with the retrieval approaches. However, fine-tuning much smaller LMs with in-context domain adaptation gives the best performances in most cases in our experiments. This phenomenon suggests we still need to fine-tune decoder-only LMs to update specific domain knowledge which will be discussed in the next section.

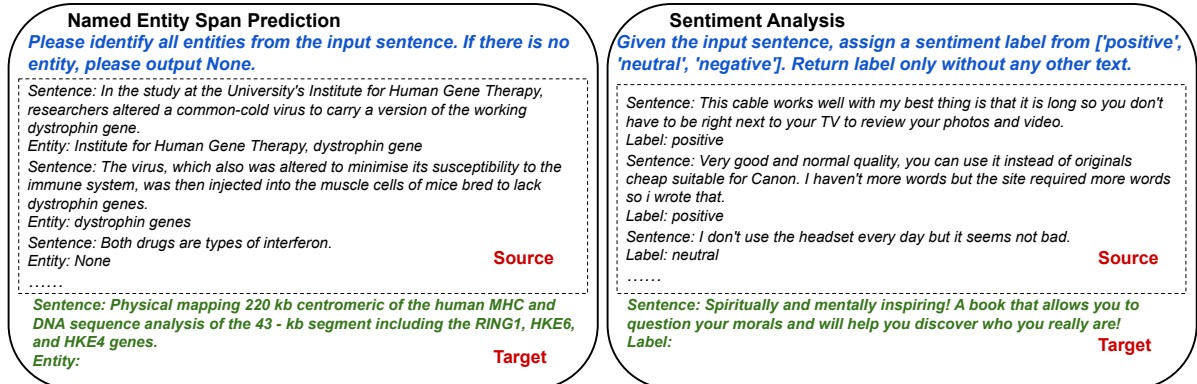

Figure 3: Examples with prompts for inference. Different from fine-tuning setting that uses target unlabeled dataset as the retrieval corpus, for inference setting, we search input-label pairs from the source labeled dataset given a target test query. Dotted boxes contain demonstrations retrieved from the source.

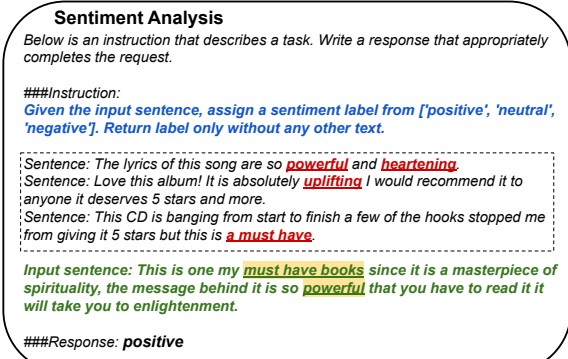

Figure 4: Illustration of crafted training example $[\text{prompt}; \overline{x}^T; x^S; y]$, dotted box contains $k = 3$ input-only demonstrations from target unlabeled dataset.

### 3.3.2 Fine-tuning

In this work, we fine-tune LLaMA (Touvron et al., 2023) with a parameter efficient approach, i.e., Low-Rank Adaptation (LoRA) (Hu et al., 2021). LoRA maintains the weights of pre-trained LMs while introducing trainable rank decomposition matrices into each transformer layer, making it feasible to fine-tune larger LMs with much fewer computational resources [2].

Similar to the method proposed in Section 3.2, we first retrieve top-k contexts from the target unlabeled set, given a source input query. We then insert these contexts in between the instruction and the source input sentence[3] (see an example in Fig. 4). Next, we finetune the decoder-only LMs given the crafted example $[\text{prompt}; \overline{x}^T; x^S; y] = [t_0, t_1, t_2, \cdots]$ and the source label. Specifically, with the Casual Language Modeling (CLM) mech-

anism, the objective is to predict the next token $t_i$:

$$\min_\theta \sum_i -\log \Pr_\theta(t_i|t_0, t_1, \cdots, t_{i-1}). \quad (5)$$

Different from section 3.2, for decoder-only LMs, the retrieved target contexts $\overline{x}^T$ need to be positioned before the source input $x^S$ as the model will learn in an autoregressive manner. Moreover, instead of only calculating token prediction loss on the response/output $y$ which is adopted for the In-context Task Learning objective as discussed in Section 3.1, we propose to compute the loss on every token within $[\text{prompt}; \overline{x}^T; x^S; y]$. Objective (5) can be decomposed into two objectives: 1) When $t_i \in \overline{x}^T$, the loss corresponds to token predictions in the target domain, analogous to the in-context language modeling objective; 2) When $t_i \in y$, the loss relates to in-context task learning which aims to generate task label given both target contexts and the source input. The objective (5) thus merges two proposed in-context objectives into a unified function.

## 4 Experiments

### 4.1 Datasets

**NER datasets** We experiment on 7 NER datasets covering four domains: News, Social media, Financial, and Biomedical. Under the **News** domain, CoNLL-03 English dataset (Sang and De Meulder, 2003) is the most popular NER dataset, and we treat it as the source domain dataset. The other three domains serve as target domains. For the **Social Media** domain, we use WNUT-16 (Strauss et al., 2016) and WNUT-17 (Derczynski et al., 2017) collected from Twitter. For the **Financial** domain, we use

---

[2] In our experiment, trainable parameters only account for 0.24% of the entire LLaMA-7B parameters.

[3] We follow the template from Standford Alpaca (Taori et al., 2023)

FIN (Alvarado et al., 2015) which is a dataset of financial agreements. For the **Biomedical** domain: we use BC2GM (Smith et al., 2008), BioNLP09 (Kim et al., 2009), and BC5CDR (Li et al., 2016). Note that for different domains, entity types are different. For unsupervised domain adaptation, to ensure source and target domains share the same label space, we remove the entity types and convert all label formats to the BIO scheme[4], similar to the problem of entity span prediction.

**Sentiment Analysis datasets** We use the Amazon review dataset (He et al., 2018) which contains four domains: Book (BK), Electronics (E), Beauty (BT), and Music (M). The original crawled reviews contain star ratings (1 to 5 stars). Following previous work (He et al., 2018; Ye et al., 2020), we label them with rating $< 3, > 3, = 3$ as negative, positive, and neutral respectively. There are in total 12 adaptation scenarios, and we select 6 of them in our experiment.

Statistics and the data splits of all the datasets can be found in Appendix A.

## 4.2 Experiment Configurations

For our retrieval system, we use SimCSE Roberta-Large (Gao et al., 2021) trained on NLI datasets[5] as the retrieval model for the SA task, and use RoBERTa-large (Liu et al., 2019) for BERTScore (Zhang et al., 2020) for the NER task[6]. We set $k = 5$ for top-$k$ retrieval from the target domain. For the encoder-only model, we select XLM-RoBERTa-large (Conneau et al., 2020) as a basis which has 561M parameters. For the decoder-only model, we use LLaMA-7B[7] (Touvron et al., 2023) and fine-tune it with LoRA (Hu et al., 2021). For inference-only LMs, we choose Chat-GPT and LLaMA-Alpaca[8]. For ChatGPT, we use gpt-3.5-turbo[9]. LLaMA-Alpaca uses LoRA to fine-tune the LLaMA-7B model on Alpaca (Taori et al., 2023) dataset which is generated with Self-Instruct (Wang et al., 2022).

## 4.3 Implementation Details

For training the RoBERTa model, we fine-tune contextualized embeddings using AdamW (Kingma

---

and Ba, 2015; Loshchilov and Hutter, 2018). In the experiments on NER datasets, the learning rate is set to 1e-5 for RoBERTa and 0.05 for CRF. For SA datasets, we set the learning rate to 5e-5 and use a linear scheduler with warm-up steps 10% of the total training steps. The weight factor $\lambda$ in (4) equals to 0.2.

[4]For example, entity "Los Angeles" has the label "B-LOC I-LOC", we convert it to "B I".

[5]https://github.com/princeton-nlp/SimCSE

[6]https://github.com/Tiiiger/bert_score

[7]For computing efficiency, we adopt the 7B model, but our method can be easily extended to larger LM families.

[8]https://huggingface.co/tloen/alpaca-lora-7b

[9]May 24 version of ChatGPT is used for experiments.

For LLaMA-LoRA, we set the rank $r$ to be 16, dropout rate to be 0.05. Trainable parameters only account for $0.24\%$ of the entire LLaMA-7B parameters. We fine-tune LLaMA-LoRA with batch size 256, learning rate 3e-4, and train 5 epochs with early stopping. With the help of LoRA, each adaptation scenario only requires less than 1 hour of training time on a single A100 GPU.

## 4.4 Results

We experiment with the following settings and baselines. For Inference-only experiments:

**No demo** performs zero-shot inference on each target test input without demonstrations.

**Rand demo** samples demonstrations randomly from the source domain.

**Retr demo** retrieves top-5 demonstrations from the source domain, corresponding to the approach mentioned in Section 3.3.1.

For fine-tuning experiments:

**No-ICL** does not retrieve any target context for each source input. The model is only trained on source inputs.

**ICL-rand** investigates the effectiveness of the task objective (1). Instead of retrieval, we randomly sample contexts from the target domain. In this case, the model is not exposed to semantically similar contexts in the target domain to enhance knowledge transfer via (1).

**ICL-sup** only trains the model via the task objective (1). This investigates the effectiveness of the language modeling objective (2). For the encoder-only model, we do not mask any token. For the decoder-only model, we calculate the loss corresponding to the response/output positions.

**ICL-source** further investigates the effectiveness of the target contexts. Here we retrieve contexts solely from the source domain instead of the target domain. Hence, the model learns to perform the task and language modeling within the source distribution.

**DAICL** is our proposed method domain-adaptive in-context learning as shown in Section 3.2 and Section 3.3.2. As described in Section 3.1, this

| | | Financial | Social Media | | Biomedical | | |
|---|---|---|---|---|---|---|---|
| | | FIN | WNUT-16 | WNUT-17 | BC2GM | BioNLP09 | BC5CDR |
| **Inference only** | | | | | | | |
| LLaMA-Alpaca | No demo | 16.69 | 14.71 | 16.20 | 13.74 | 16.44 | 21.64 |
| | Rand demo | 13.59 | 23.20 | 22.10 | 20.69 | **24.46** | 25.83 |
| | Retr demo | **20.18** | **29.5** | **26.73** | **22.56** | 22.98 | **27.26** |
| ChatGPT | No demo | 19.60 | 32.10 | 33.45 | 19.90 | 15.44 | 37.16 |
| | Rand demo | **20.82** | **39.73** | **39.45** | **26.92** | **21.71** | **37.85** |
| | Retr demo | 19.88 | 38.28 | 38.10 | 24.17 | 18.98 | 35.71 |
| **Fine-tuning** | | | | | | | |
| RoBERTa | Vu et al. (2020) | 23.38 | 47.11 | – | 30.81 | 29.24 | – |
| | No-ICL | $24.17_{1.3}$ | $68.49_{0.2}$ | $63.18_{0.3}$ | $27.69_{1.6}$ | $33.67_{1.1}$ | $21.84_{2.2}$ |
| | ICL-rand | $24.91_{2.9}$ | $69.26^{\dagger}_{0.6}$ | $64.66^{\dagger}_{0.3}$ | $30.68^{\dagger}_{1.6}$ | $35.19^{\dagger}_{0.9}$ | $26.93^{\dagger}_{1.9}$ |
| | ICL-sup | $26.24^{\dagger}_{2.1}$ | $70.89^{\dagger}_{0.4}$ | $65.40^{\dagger}_{0.4}$ | $28.07_{1.4}$ | $34.11_{0.9}$ | $23.20^{\dagger}_{2.2}$ |
| | ICL-source | $24.91_{1.2}$ | $68.84_{0.3}$ | $63.38_{0.2}$ | $26.96_{1.8}$ | $32.07_{1.2}$ | $22.06_{2.0}$ |
| | **DAICL** | $\mathbf{27.22}^{\dagger}_{2.1}$ | $\mathbf{71.79}^{\dagger}_{0.4}$ | $\mathbf{65.79}^{\dagger}_{0.2}$ | $\mathbf{32.51}^{\dagger}_{1.1}$ | $\mathbf{36.81}^{\dagger}_{0.6}$ | $25.92^{\dagger}_{1.8}$ |
| LLaMA-LoRA | No-ICL | 15.20 | 45.22 | 53.92 | 24.24 | **26.29** | 25.35 |
| | ICL-rand | 12.68 | 42.09 | 51.08 | 23.06 | 21.66 | 21.28 |
| | ICL-sup | **15.81** | 45.70 | 54.32 | 24.83 | 25.00 | 26.91 |
| | ICL-source | 14.73 | 45.30 | 53.29 | 24.54 | 23.92 | 24.96 |
| | **DAICL** | 14.82 | **46.51** | **55.08** | **26.02** | 24.21 | **28.96** |

Table 1: F1 results of Named Entity Span prediction tasks. The source domain is the CoNLL-03 dataset, and the target domains are financial, social media, and biomedical. For RoBERTa, results are reported with average and standard deviation in 5 runs, † represents the model is significantly stronger than the baseline model No-ICL with $p < 0.05$. For LLaMA, due to the cost of inference computation, we only perform a single run.

method retrieves related contexts from the target domain and combines two objectives to perform domain-adaptive ICL.

The experiment results for NER and SA are illustrated in Table 1 and Table 2, respectively. Below we conclude with some interesting findings.

**Adaptive ICL benefits UDA by learning two objectives simultaneously.** Given the results in Table 1 and Table 2, we can observe that our proposed method DAICL which learns two objectives simultaneously surpasses baselines with a large margin in most adaptation scenarios. From the result of ICL-sup, we find that training with the task objective alone could slightly help UDA. As discussed in Section 3, the benefit originates from incorporating the target contexts for task discrimination. By comparing DAICL with ICL-sup and ICL-source, we can conclude that the proposed in-context adaptation strategy enhances domain adaptation by jointly learning the task signal and language modeling simultaneously.

**Retrieving OOD examples could be disappointing for LLMs.** From the RoBERTa results of ICL-rand, we find that random target contexts can improve NER (compared with No-ICL) by a small margin. One possible reason is that random contexts from the target domain could still encourage the model to learn the target distribution via (2).

However, ICL-rand significantly impedes the performance of Sentiment Analysis. We conjecture that ICL-rand might select target contexts with opposite sentiment labels from the source input, negatively affecting the learning process.

Surprisingly, ChatGPT with random out-of-distribution (OOD) demonstrations achieves higher scores than retrieval in all NER and SA experiments (Rand demo vs. Retr demo). Previous work reveals that choosing demonstration examples that are close to the test input significantly enhances the effectiveness of ICL (Liu et al., 2022; Rubin et al., 2022). However, they retrieve from a labeled training set in which the distributions of the text and label space are identical with the test input. In contrast, in transfer setting which is close to the real-world scenario, we only have OOD input-label pairs from another labeled dataset. We make a hypothesis regarding this observation, for cross-domain ICL, providing diverse and distinct OOD demonstrations is more beneficial for LLMs to understand the task and generalize.

**Fine-tuning is still beneficial for UDA.** Under the UDA setting where labels only exist in the source domain, we can prompt LLMs with input-label pairs from the source domain to infer the target label (inference-only). Another option is to fine-tune smaller LMs to adapt task-aware knowledge from the source to the target domains. A natural ques-

| | | E→BK | BT→BK | BK→BT | BK→M | BK→E | M→BT | Ave. |
|---|---|---|---|---|---|---|---|---|
| | | **Inference only** | | | | | | |
| LLaMA-Alpaca | No demo | **61.53** | 61.53 | 63.72 | 58.86 | 59.41 | 63.72 | 61.46 |
| | Rand demo | 54.33 | 55.45 | 60.48 | 49.09 | 51.98 | 63.78 | 55.85 |
| | Retr demo | 60.9 | **63.58** | **69.35** | **60.33** | **61.36** | **67.82** | **64.06** |
| ChatGPT | No demo | 72.68 | 72.68 | 72.27 | 70.06 | 69.83 | 72.27 | 71.63 |
| | Rand demo | **73.10** | **73.27** | **74.37** | **71.18** | **71.44** | **74.3** | **72.94** |
| | Retr demo | 73.07 | 71.92 | 73.82 | 69.69 | 71.00 | 73.57 | 72.18 |
| | | **Fine-tuning** | | | | | | |
| RoBERTa | Long et al. (2022) | $70.33_{0.3}$ | $70.92_{0.6}$ | $64.13_{1.4}$ | $64.67_{1.7}$ | $62.36_{0.7}$ | $65,40_{0.8}$ | 66.30 |
| | Ye et al. (2020) | $70.90_{0.4}$ | $71.38_{0.8}$ | $67.48_{0.4}$ | $\mathbf{67.16}_{0.6}$ | $64.00_{1.2}$ | $70.71_{0.3}$ | 68.61 |
| | No-ICL | $68.33_{0.5}$ | $69.85_{0.6}$ | $65.92_{1.1}$ | $61.47_{1.7}$ | $61.36_{0.7}$ | $67.43_{0.8}$ | 65.73 |
| | ICL-rand | $67.61_{0.8}$ | $68.74_{0.6}$ | $64.80_{1.3}$ | $61.59_{1.9}$ | $61.44_{0.9}$ | $66.72_{1.7}$ | 65.15 |
| | ICL-sup | $69.68^{\dagger}_{0.6}$ | $71.15^{\dagger}_{0.5}$ | $\mathbf{68.79}^{\dagger}_{1.4}$ | $64.88^{\dagger}_{1.1}$ | $63.16^{\dagger}_{1.0}$ | $69.15^{\dagger}_{1.1}$ | 67.80 |
| | ICL-source | $68.70_{0.8}$ | $70.64^{\dagger}_{0.8}$ | $65.29_{1.4}$ | $61.81_{2.2}$ | $61.75_{1.4}$ | $66.89_{1.9}$ | 65.84 |
| | **DAICL** | $71.21^{\dagger}_{0.5}$ | $72.81^{\dagger}_{0.9}$ | $68.64^{\dagger}_{1.7}$ | $66.93^{\dagger}_{0.8}$ | $66.08^{\dagger}_{0.7}$ | $71.44^{\dagger}_{0.9}$ | 69.52 |
| LLaMA-LoRA | No-ICL | 74.15 | 74.30 | 72.97 | 70.42 | 70.08 | 70.15 | 72.01 |
| | ICL-rand | 65.22 | 64.17 | 60.48 | 61.95 | 59.36 | 63.44 | 62.43 |
| | ICL-sup | 76.10 | 75.20 | 72.25 | 71.63 | **71.78** | 70.54 | 72.75 |
| | ICL-source | 70.18 | 68.45 | 68.46 | 63.27 | 67.23 | 67.94 | 67.59 |
| | **DAICL** | **77.30** | **76.30** | **74.02** | **73.40** | 70.38 | **72.37** | **74.13** |

Table 2: Accuracy(%) results of Amazon Review Sentiment Analysis. For example, E→BK represents training on Electronics (E) and adapting to Book (BK). There are 4 domains available, we choose 6 out of 12 adaptation tasks.

tion to ask is can the few-shot prompting paradigm substitute the fine-tuning paradigm? In NER experiments, ChatGPT achieves very low performances, but fine-tuning a much smaller RoBERTa model achieves state-of-the-art scores in most adaptation scenarios. In SA experiments, fine-tuning LLaMA with even fewer trainable parameters (1.7M) outperforms all the other methods. Hence, we hypothesize that although LLMs have strong generalization ability, they cannot tackle problems in all domains. When it comes to UDA, designing an effective adaptation strategy is still beneficial.

## 4.5 Analysis

**Adaptive ICL or Adaptive Pre-training?** In Section 3.1, we propose to learn the two objectives simultaneously with the help of the target contexts. What if we separate the two objectives into different training stages? In the first stage, we continue pre-training LMs on the unlabeled target domain dataset with the language modeling objective. In the second stage, supervised fine-tuning is performed on the labeled source domain dataset with the task objective. This two-step UDA procedure is called adaptive pre-training or post pre-training (Han and Eisenstein, 2019; Vu et al., 2020; Karouzos et al., 2021). There are two differences between adaptive pre-training and adaptive ICL which we propose: 1) adaptive ICL mixes a source input with target contexts when performing task predictions while adaptive pre-training only takes

the source input; 2) adaptive ICL learns two losses simultaneously, and for decoder-only model, we only have one type of task which merges these two losses intrinsically.

| | WNUT17 | BC2GM | E→BK | M→BT |
|---|---|---|---|---|
| pre-train | 54.62 | 25.78 | 74.20 | 70.45 |
| No-ICL | 53.92 | 24.24 | 74.15 | 70.15 |
| DAICL | **55.08** | **26.02** | **77.30** | **72.37** |

Table 3: A comparison of adaptive ICL and adaptive pre-training for LLaMA. On NER, we use CoNLL-03→WNUT17 and CoNLL-03→BC2GM. For SA, we use E→BK and M→BT.

To compare the two approaches, we conduct experiments on LLaMA-LoRA to perform adaptive pre-training. In the first stage, we pre-train LoRA weights using target unlabeled texts. In the second stage, we start from the LoRA checkpoint obtained in the previous stage and continue fine-tuning it with task supervision. We use the same Alpaca template but do not provide demonstrative context. Results can be found in Table 3. No ICL is identical to the second stage in adaptive pre-training.

We could observe that pre-training only gains marginal benefits for SA tasks compared with No-ICL. We conjecture that the domain gap is smaller in SA datasets than in NER datasets. The proposed adaptive ICL strategy outperforms adaptive pre-training, which could be attributed to the fact that the decoder-only model under adaptive ICL can learn the two objectives with demonstrative contexts.

## 5 Related Work

**Unsupervised Domain Adaptation**

Traditional methods include Pseudo-labeling (Ye et al., 2020), Pivot-based approach (Pan et al., 2010), and adversarial neural network (Ganin et al., 2016). Recently, Adaptive pre-training on domain-specific corpora has proven to be an effective process for adaptation, such as BioBERT (Lee et al., 2019) which is a specialized variant of BERT. Han and Eisenstein (2019) proposes AdaptaBERT, which includes a second phase of unsupervised pre-training for BERT in unsupervised domain adaptation. Karouzos et al. (2021) proposes a mixed multi-task loss to learn classification and MLM. Chronopoulou et al. (2019) utilizes an auxiliary LM loss to prevent catastrophic forgetting in transfer learning.

**Retrieval-Augmented Language Models**

Retrieval-based LMs have shown to be effective in improving LM performance (Asai et al., 2023). The retriever with various knowledge datastores can provide up-to-date information since LMs cannot memorize all long-tail knowledge in the parameters. REALM (Guu et al., 2020) pre-trains and fine-tunes an encoder-only model jointly with a knowledge retriever by modeling documents as latent variables and marginalizing over all possible documents. While RAG (Lewis et al., 2020) fine-tunes an encoder-decoder model with a non-parametric retriever by fixing the search index. Atlas (Izacard et al., 2022) combines RAG with pre-training on open-domain QA and knowledge-intensive tasks. Replug (Shi et al., 2023) proposes adapting the dense retriever to the black-box large language models to reduce the generating perplexity.

**In-Context Learning**

In the context of ICL, previous studies indicate that it primarily exposes the model's infrastructure learned during pre-training. Xie et al. (2022) provides evidence that ICL can be interpreted as a type of Bayesian inference, where demonstrations act as noisy evidence. (Min et al., 2022) shows that the advantages of ICL mainly stem from having the appropriate distribution of inputs and labels, rather than solely focusing on the correctness of individual labels. Previous research has revealed that in scenarios where abundant training data is accessible, retrieving examples that are similar to the test input as demonstrations significantly enhances ICL performance. Liu et al. (2022) introduces a retrieval module for GPT-3 (Brown et al., 2020)

and they also fine-tune the retrieval model, leading to stronger ICL performance. Rubin et al. (2022) trains a dense retriever to select demonstrations that have a positive impact on the learning process.

## 6 Conclusion

In this work, we propose domain-adaptive in-context learning to acquire knowledge of both the target domain distribution and the discriminative task signal simultaneously. We develop different prompting and fine-tuning strategies that take into account various LM architectures and different language modeling mechanisms. Overall, our framework demonstrates significant performance gains over an extensive spectrum of cross-domain experiments, and we perceive that fine-tuning is still effective and promising in the era of large language models when it involves domain shift.

## 7 Limitations

Our retrieval system is based on SimCSE and BERTScore to choose semantically similar contexts following previous work. However, we do not explore other scoring and re-ranking metrics, or explore methods to train a dense retriever. On the other hand, it is hard to tell what makes a good demonstration simply based on a retrieval system, considering that the retrieval system does not have access to the inference task. We leave this for future work to explore what is a good demonstrative example when encountering domain shift.

## 8 Ethics Statement

To ensure the ethical use of Artificial Intelligence in the legal field, we have taken measures such as anonymizing sensitive information in real-world datasets. In addition, our model's predictions should be served as supportive references for judges, assisting them in making judgments more efficiently, rather than solely determining the judgments.

## Acknowledgements

This work is partially supported by the 2020 Microsoft Research Asia collaborative research grant. Sinno J. Pan thanks for the support from HK Global STEM Professorship and the JC STEM Lab of Machine Learning and Symbolic Reasoning.

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

## A Datasets

|          | # Train | # Dev | # Test |
|----------|---------|-------|--------|
| CoNLL-03 | 14,987  | 3,466 | 3,684  |
| FIN      | 1169    | –     | 306    |
| WNUT-16  | 2,394   | 1,000 | 3,849  |
| WNUT-17  | 3,394   | 1,009 | 1,287  |
| BC2GM    | 12574   | 2519  | 5038   |
| BioNLP09 | 7462    | 1448  | 2446   |
| BC5CDR   | 4,560   | 4,581 | 4,797  |

Table 4: Statistics of the dataset split of NER dataset.

For NER datasets, we select CoNLL-03 training as the source labeled dataset and a CoNLL-03 dev set as the validation set for adaptation. When adapting to a target domain, for example, WNUT16, we use WNUT16 training set as the unlabeled target dataset by discarding all labels from this training set. That is, in our approach, for the fine-tuning setting, we retrieve text-only examples from WNUT16 training dataset as the contexts of source input CoNLL03. Statistics can be found in Table 4

| DOMAIN |       | # Neg | # Neu | # Pos | Total |
|--------|-------|-------|-------|-------|-------|
| BOOK   | Set 1 | 2000  | 2000  | 2000  | 6000  |
|        | Set 2 | 513   | 663   | 4824  | 6000  |
| ELEC   | Set 1 | 2000  | 2000  | 2000  | 6000  |
|        | Set 2 | 694   | 489   | 4817  | 6000  |
| BEAUTY | Set 1 | 2000  | 2000  | 2000  | 6000  |
|        | Set 2 | 616   | 675   | 4709  | 6000  |
| MUSIC  | Set 1 | 2000  | 2000  | 2000  | 6000  |
|        | Set 2 | 785   | 774   | 4441  | 6000  |

Table 5: Statistics of the dateset split of NER dataset.

For the Amazon review dataset, it does not remove neutral labels, which is advantageous in unsupervised domain adaptation (UDA) scenarios where label information from the target domain is unavailable. A summary of this dataset can be found in Table 5. For SA, each dataset consists of two sets. Set 1 contains 6,000 instances with balanced class labels, while Set 2 comprises instances randomly sampled from a larger dataset (McAuley et al., 2015), preserving the authentic label distribution. It is important to note that there is no overlap between the examples in these two sets. Following the approach outlined in (He et al., 2018), Set 1 is used as the training set for the source domain. While the label distribution in the target domain is unpredictable and beyond our control in real-life scenarios, it is more reasonable to use Set 2 as the unlabeled set for the target domain. Finally, the model is evaluated on Set 1 from the target domain. Regarding the data split, a validation set is created

by randomly sampling 1000 instances from the source labeled Set 1. For example, when performing E→BK adaptation task, we use Electronics Set 1 as the training set and validation set, we use Book Set 2 as the target unlabeled set, and we retrieve similar examples from this set as contexts. The evaluation will be performed in Book Set 1.

## B Example Input and Output Pairs of ChatGPT and LLaMA

Table 6: Example input and output pairs for ChatGPT on NER dataset BC2GM. No demonstration.

---

Please identify all entities from the input sentence. If there is no entity, please output None.

**Sentence:** Physical mapping 220 kb centromeric of the human MHC and DNA sequence analysis of the 43 - kb segment including the RING1 , HKE6 , and HKE4 genes .
**Entity:** - Physical mapping
- human MHC
- DNA sequence analysis
- RING1 gene
- HKE6 gene
- HKE4 gene

Please identify all entities from the input sentence. If there is no entity, please output None.

**Sentence:** DNA elements recognizing NF - Y and Sp1 regulate the human multidrug - resistance gene promoter .
**Entity:**
1. DNA elements
2. NF-Y
3. Sp1
4. human multidrug-resistance gene promoter

Please identify all entities from the input sentence. If there is no entity, please output None.

**Sentence:** Like other IAPs , ch - IAP1 contains the N - terminal baculovirus IAP repeats and C - terminal RING finger motifs .
**Entity:**
IAPs, ch-IAP1, baculovirus IAP repeats, RING finger motifs.

Please identify all entities from the input sentence. If there is no entity, please output None.

**Sentence:** To clarify the difference , both the Crk II and Crk II - 23 , proteins were expressed in E . coli and examined their binding capacity in vitro .
**Entity:**
- Crk II
- Crk II - 23
- E. coli

Please identify all entities from the input sentence. If there is no entity, please output None.

**Sentence:** A GT - rich sequence binding the transcription factor Sp1 is crucial for high expression of the human type VII collagen gene ( COL7A1 ) in fibroblasts and keratinocytes .
**Entity:**
1. GT-rich sequence
2. Transcription factor Sp1
3. Human type VII collagen gene (COL7A1)

Please identify all entities from the input sentence. If there is no entity, please output None.

**Sentence:** Manual sample clean - up procedures as well as the addition of an internal standard are not needed .
**Entity:**
None.

---

Table 7: Example input and output pairs for ChatGPT on NER dataset BC2GM. Retrieved demonstrations are from CoNLL03.

---

Please identify all entities from the input sentence. If there is no entity, please output None.

Sentence: In the study at the University 's Institute for Human Gene Therapy , researchers altered a common-cold virus to carry a version of the working dystrophin gene .
Entity: Institute for Human Gene Therapy
Sentence: The virus , which also was altered to minimise its susceptibility to the immune system , was then injected into the muscle cells of mice bred to lack dystrophin genes .
Entity: None
Sentence: " We agreed that following detailed scientific analysis using a methodology which would take out the maximum number of BSE cases possible .
Entity: BSE
Sentence: In the experiment , between 30 to 40 percent of the muscle fibers in one group of mice produced dystrophin for two weeks before diminishing .
Entity: None
Sentence: Stork H1 results breakdown per sector .
Entity: None

**Sentence:** Physical mapping 220 kb centromeric of the human MHC and DNA sequence analysis of the 43 - kb segment including the RING1 , HKE6 , and HKE4 genes .
**Entity:**
human MHC, RING1, HKE6, HKE4 genes.

Please identify all entities from the input sentence. If there is no entity, please output None.

Sentence: In the study at the University 's Institute for Human Gene Therapy , researchers altered a common-cold virus to carry a version of the working dystrophin gene .
Entity: Institute for Human Gene Therapy
Sentence: Individuals with the disease have a non-working version of a gene responsible for producing a crucial muscle protein called dystrophin .
Entity: None
Sentence: The virus , which also was altered to minimise its susceptibility to the immune system , was then injected into the muscle cells of mice bred to lack dystrophin genes .
Entity: BSE
Sentence: Both drugs are types of interferon .
Entity: None
Sentence: When it approved Avonex in May , the FDA said both Biogen 's product and Betaseron were developed under the incentives of the Ophran Drug Act which provides seven years of marketing exclusivity for products that treat rare diseases .
Entity: Avonex, FDA, Biogen, Betaseron, Ophran Drug Act

**Sentence:** DNA elements recognizing NF - Y and Sp1 regulate the human multidrug - resistance gene promoter .
**Entity:**
NF-Y, Sp1, human multidrug-resistance gene promoter.

Table 8: Example input and output pairs for ChatGPT on NER dataset BC2GM. Demonstrations are randomly sampled from CoNLL03.

Please identify all entities from the input sentence. If there is no entity, please output None.

Sentence: The 32-year-old defender played seven seasons with Nantes and was with Paris St Germain for five seasons .
Entity: Nantes, Paris St Germain
Sentence: 29,582
Entity: None
Sentence: The Palestinian Authority was set up under the 1993 PLO-Israel interim peace deal .
Entity: Palestinian Authority, PLO-Israel
Sentence: In Home Health said it previously recorded a reserve equal to 16 percent of all revenue related to the community liaison costs .
Entity: In Home Health
Sentence: " I realised this year , that without putting 99.9 percent of your mind into tennis , I do n't think you can successful , " said the 22-year-old Medvedev .
Entity: Medvedev

**Sentence:** Physical mapping 220 kb centromeric of the human MHC and DNA sequence analysis of the 43 - kb segment including the RING1 , HKE6 , and HKE4 genes .
**Entity:**
human MHC, RING1, HKE6, HKE4.

Please identify all entities from the input sentence. If there is no entity, please output None.

Sentence: The blue-chip CAC-40 index ended 2.43 points or 0.12 percent lower at 2,017.99 points after a brief foray into positive territory when the New York stock market opened higher .
Entity: CAC-40, New York
Sentence: American League
Entity: American League
Sentence: 1886 - At Skeleton Canyon in Arizona , Geronimo , Apache chief and leader of the last great Red Indian rebellion finally surrendered to General Nelson Miles .
Entity: Skeleton Canyon, Arizona, Geronimo, Red Indian, Nelson Miles
Sentence: ( Formula Shell leads best-of-seven series 1-0 )
Entity: Formula Shell
Sentence: PRESS DIGEST - Jordan - Aug 25 . Entity: Jordan **Sentence:** DNA elements recognizing NF - Y and Sp1 regulate the human multidrug - resistance gene promoter .
**Entity:**
NF-Y, Sp1, DNA elements.

Table 9: Example input and output pairs for LLaMA-Alpaca on NER dataset BC2GM. No demonstration. Note that Alpaca-lora is not fine-tuned on NER datasets.

---

Below is an instruction that describes a task, paired with an input that provides further context. Write a response that appropriately completes the request.

### Instruction:

Please identify all entities from the input sentence. If there is no entity, please output None.

Input sentence: Physical mapping 220 kb centromeric of the human MHC and DNA sequence analysis of the 43 - kb segment including the RING1 , HKE6 , and HKE4 genes .
### Response:
None

Below is an instruction that describes a task, paired with an input that provides further context. Write a response that appropriately completes the request.

### Instruction:

Please identify all entities from the input sentence. If there is no entity, please output None.

Input sentence: DNA elements recognizing NF - Y and Sp1 regulate the human multidrug - resistance gene promoter .
### Response:
DNA elements, NF - Y, Sp1

Below is an instruction that describes a task, paired with an input that provides further context. Write a response that appropriately completes the request.

### Instruction:

Please identify all entities from the input sentence. If there is no entity, please output None.

Input sentence: Like other IAPs , ch - IAP1 contains the N - terminal baculovirus IAP repeats and C - terminal RING finger motifs .
### Response:
None

Below is an instruction that describes a task, paired with an input that provides further context. Write a response that appropriately completes the request.

### Instruction:

Please identify all entities from the input sentence. If there is no entity, please output None.

Input sentence: A GT - rich sequence binding the transcription factor Sp1 is crucial for high expression of the human type VII collagen gene ( COL7A1 ) in fibroblasts and keratinocytes .
### Response:
human type VII collagen gene

Table 10: Example input and output pairs for our fine-tuned LLaMA-7B on NER dataset BC2GM. Inference with retrieved demonstration.

---

Below is an instruction that describes a task, paired with an input that provides further context. Write a response that appropriately completes the request.

### Instruction:

Please identify all entities from the input sentence. If there is no entity, please output None.

Sentence: In the study at the University 's Institute for Human Gene Therapy , researchers altered a common-cold virus to carry a version of the working dystrophin gene .
Entity: Institute for Human Gene Therapy
Sentence: The virus , which also was altered to minimise its susceptibility to the immune system , was then injected into the muscle cells of mice bred to lack dystrophin genes .
Entity: None
Sentence: " We agreed that following detailed scientific analysis using a methodology which would take out the maximum number of BSE cases possible .
Entity: BSE
Sentence: In the experiment , between 30 to 40 percent of the muscle fibers in one group of mice produced dystrophin for two weeks before diminishing .
Entity: None
Sentence: Stork H1 results breakdown per sector .
Entity: None

Input sentence: Physical mapping 220 kb centromeric of the human MHC and DNA sequence analysis of the 43 - kb segment including the RING1 , HKE6 , and HKE4 genes .
### Response:
Human MHC, RING1, HKE6, HKE4 genes

Below is an instruction that describes a task, paired with an input that provides further context. Write a response that appropriately completes the request.

### Instruction:

Please identify all entities from the input sentence. If there is no entity, please output None.

Sentence: In the study at the University 's Institute for Human Gene Therapy , researchers altered a common-cold virus to carry a version of the working dystrophin gene .
Entity: Institute for Human Gene Therapy
Sentence: Individuals with the disease have a non-working version of a gene responsible for producing a crucial muscle protein called dystrophin .
Entity: None
Sentence: The virus , which also was altered to minimise its susceptibility to the immune system , was then injected into the muscle cells of mice bred to lack dystrophin genes .
Entity: BSE
Sentence: Both drugs are types of interferon .
Entity: None
Sentence: When it approved Avonex in May , the FDA said both Biogen 's product and Betaseron were developed under the incentives of the Ophran Drug Act which provides seven years of marketing exclusivity for products that treat rare diseases .
Entity: Avonex, FDA, Biogen, Betaseron, Ophran Drug Act

Input sentence: DNA elements recognizing NF - Y and Sp1 regulate the human multidrug - resistance gene promoter .
### Response:
NF - Y, Sp1