# OpenReview forum: "Adapt in Contexts: Retrieval-Augmented Domain Adaptation via In-Context Learning"
_EMNLP/2023/Conference — EMNLP 2023 Main_

### Official Review · Reviewer_PvYw · 2023-08-02

**Soundness:** 2

**Excitement:**

2: Mediocre: This paper makes marginal contributions (vs non-contemporaneous work), so I would rather not see it in the conference.

**Paper Topic And Main Contributions:**

This paper conducted research to improve the performance of domain adaptation using in-context learning. Specifically, the proposed method involves finding and utilizing samples similar to the input through the process.
Firstly, the input sentence and k semantically similar sentences are retrieved using third-party models (e.g., SimCSE). Next, the retrieved sentences are combined with the target input, and two types of losses are utilized: the basic loss for label prediction and the language modeling loss. The model was experimented with in NER and sentiment analysis research, and it demonstrated better performance compared to using the conventional ICL method.

**Reasons To Accept:**

- Conducted rigorous experiments using diverse language models with varying methods.
- The proposed methodology generally demonstrates better results compared to simply adding random samples from various datasets.

**Reasons To Reject:**

- First of all, I am not entirely sure if large language models (LLMs) are suitable for domain adaptation research. Domain adaptation research aims to maintain the performance achieved by training on a source dataset when applied to a target distribution. However, when using LLMs only for inference, there are some issues. LLMs lack a clear criterion to define in-distribution (source and target distribution). Models like ChatGPT and LLAMA are already considered general-purpose language models, assuming they have been trained to some extent on both source and target domains. Therefore, when utilizing LLMs for domain adaptation, more thoughtful or convincing justifications are required rather than relying on methods that distinguish source and target as in traditional domain adaptation using smaller models. Similarly, conducting experiments with LLMs for inference only, without training them, appears irrelevant to domain adaptation research since there is no explicit learning for the source distribution.

- The methodology lacks novelty and significance. Utilizing basic loss functions for label prediction and language modeling, it does not present an exceptionally innovative approach, as similar methods have been attempted in various tasks before. Furthermore, the proposed method does not seem to demonstrate significant differences when compared to using these methods alone without incorporating ICL. While the proposed approach comes with some noticeable drawbacks, such as long input length and an additional third-party retrieval module, its merits appear to be limited in comparison.

**Reproducibility:**

3: Could reproduce the results with some difficulty. The settings of parameters are underspecified or subjectively determined; the training/evaluation data are not widely available.

**Reviewer Confidence:**

4: Quite sure. I tried to check the important points carefully. It's unlikely, though conceivable, that I missed something that should affect my ratings.

---

> ### Author Rebuttal · Authors · 2023-08-29
>
> Thanks for your valuable comments!
>
> #### **comment #1**:
> LLMs are not suitable for domain adaptation research:
> - (1) LLMs lack a clear criterion to define in-distribution, we should not rely on criterion that distinguish the source and target as in traditional UDA using SLMs.
> - (2) Conducting inference experiments with LLMs is irrelevant to domain adaptation.
>
> #### **response #1**:
> We think domain related problem is still worth studying in the era of LLMs. Below are our justifications:
>
> - (1) **We focus on fine-tuning instead of inference-only.** Unlike what the reviewer mentioned, inference-only experiments with LLMs are not the focus of this paper. Our main contribution lies in providing a unified framework with effective prompting and finetuning strategies for in-context domain adaptation accounting for different LM architectures. The reason for conducting inference-only experiments is to prove that fine-tunig is still beneficial for unsupervised domain adaptation compared with inference-only LLMs. In NER experiments, ChatGPT achieves very low performances, but fine-tuning a much smaller RoBERTa model achieves SOTA scores in most adaptation scenarios. In Sentiment Analysis experiments, fine-tuning LLaMA with even fewer trainable parameters (1.7M, Trainable parameters only account for 0.24\% of the entire LLaMA-7B parameters) outperforms all the other methods. Hence, we hypothesize that even though LLMs have strong generalization ability, they cannot tackle problems in all domains. When it comes to UDA, designing an effective adaptation strategy (fine-tuning) is still promising, which is the goal of this work.
>
> - (2) **We can still distinguish source and target domains for inference using LLMs.** LLMs usually take a few input-output demonstrations as context (few-shot) to perform a task. However, in most real-life scenarios, we do not have such labeled in-distribution dataset (target domain, e.g., biomedical) to offer input-output demonstrations. As a result, we need to resort to other labeled datasets (source domain, e.g., social media) to construct those demonstrations. This has demonstrated better performances than not providing any demonstrations (zero-shot) as shown in Table 1 and 2. In such cases, the demonstrations (few-shot) come from the source domain and the target input to evaluate comes from the target domain.
>
> - (3) **Out-of-distribution demonstrations is worth studying for LLMs inference-only setting.** Since in most practical circumstances, we draw demonstrations from another labeled dataset with a different text distribution (e.g., changing topics and genres), thus we suffer from the out-of-distribution problem. Why this is worth studying? Previous studies [1] have shown that retrieval is always helpful under a single domain by selecting input-output demonstrations from the labeled in-distribution corpus (the train set). However, from our ChatGPT results, retrieval under the out-of-distribution setting fails, compared to randomly-sampled demonstrations. We make a hypothesis that providing diverse and distinct examples is more beneficial for LLMs to understand the task and generalize under the domain adaptation setting. While further in-depth experiments and studies are worthwhile studying under the OOD setting.
>
> - (4) **LLMs still face challenges with certain domains.** LLMs pretrianed over a huge corpus demonstrate generlizabilities towards a diverse set of tasks. They can be viewed as mastering fundamental knowledge, but that does not necessarily indicate they can perform well on any domain. There still exist many domains which are not frequently seen in the pretrained corpus, especially under real scenarios considering company-owned datasets. Hence, it is still crucial and necessary to investigate domain adaptation in the era of LLMs.
>
> #### **comment #2**:
> - Utilizing basis loss functions for label prediction and language modeling is similar to previous works, similar methods have been attempted in various task before.
> - And the proposed method does not seem to demonstrate significant differences when comparing to these previous works, while the proposed approach comes with some noticeable drawbacks, such as long input length and an additional third-party retrieval module.
>
> #### **response #2**:
> We provide the following justifications to address some misunderstandings:
> - First, **our method is not a simple utilization of two basis losses.** we present our methods with the discussion of in-context learning, we emphasize the idea of learning a model with the retrieved semantically-rich and demonstrative contexts. Besides, our approach is different from previous approaches (domain adaptive pretraining), which learns these two objectives in two steps: first, pretraining on the target domain with MLM objective, second, supervised training on the source domain. In contrast, our proposed adaptive ICL learns the task discrimination and target distribution simultaneously with an in-context manner, and we learn target distribution from a source-target joint view (we mix the source and target as context, the model learns to fuse the distribution and bridge the domain gap). Yet the previous works only pretrain from a single view (target only). It would be great if the reviewer could provide us some references that you think is similar to ours.
>
> - Second, in our paper, we present a unified framework with efficient prompting and fine-tuning strategies accounting for different LM architectures. **Specifically for the decoder-only model (LLaMA), our method is very different from previous works (encoder-only models, BERT).**
>
> - Third, in our experiment, **we have demonstrated that our method can surpass the baselines with a large margin.** We also conduct analysing experiments to compare our methods with the previous language modeling based UDA methods. We discuss the detailed differences between the two methods. We conduct experiments on LLaMA and find that the proposed adaptive ICL outperforms adaptive pretraining.
>
> - Fourth, **input length and the additional retrieval module should not be considered as drawbacks.** In the NLP community, both in-context learning and retrieval augmentation have garnered significant attention and discussion due to their potential to provide semantically-enriched contexts and exceptional performances. And it has become a common practice to use in-context demonstrations with the help of a retrieval module on various applications. Therefore we believe that exploring these aspects enhances the scope of domain adaptation and aligns with current trends in the field. Besides, we would like to clarify the effectiveness of our retrieval module. Given a corpus, building the indexes and search for top-k relevant sentences for 1k instances cost less than two minutes.
>
> Thanks for your comments again!
>
> [1] What Makes Good In-Context Examples for GPT-3? Jiachang Liu, Dinghan Shen, Yizhe Zhang, Bill Dolan, Lawrence Carin, Weizhu Chen

---

### Official Review · Reviewer_gjmy · 2023-08-06

**Soundness:** 4

**Excitement:**

4: Strong: This paper deepens the understanding of some phenomenon or lowers the barriers to an existing research direction.

**Paper Topic And Main Contributions:**

The paper proposes methods for unsupervised domain adaptation of language models via in-context learning. Different methods are developed for encoder-only and decoder-only method under an inference-only and a fine-tuning setup. The main idea under the inference-only setup is to augment target unlabeled examples with retrieved demonstrations from the source domain. The main idea under the fine-tuning setup is to augment source examples with semantically similar (unlabeled) examples from the target domain and fine-tune the model with language modeling and in-context task learning objectives. The proposed methods are evaluated on several datasets for NER and sentiment analysis.

**Questions For The Authors:**

Even for encode-only models the target example could be placed before the source example in the prompt. This could help reduce the gap between the solutions for encoder-only and decoder-only models. Did the authors consider this?

In decoder-only training (Eq. 5) would it help to weight  the prediction of $y$ higher than the prediction of $x^T$? In other words, weigh the relative importance of the supervised task objective vs. LM objective?

LLaMA is less effective than RoBERTa in several cases. Could the authors give practical guidelines on which model & ICL technique to employ for each model class?

Are the authors planning to share their code?

**Reasons To Accept:**

The paper shows that large language models, including ChatGPT have subpar performance, especially on NER and demonstrates practical ways to boost their accuracy both in inference-only and fine-tuning setups. Extensive experiments across datasets demonstrate the effectiveness of exemplar retrieval and further fine-tuning when possible.

**Reasons To Reject:**

It is not clear which part of the paper's first claimed contribution ("To the best of our knowledge, we take the first step towards in-context domain adaptation, which takes advantage of LM architectures to learn target domain distribution while adapting in contexts") is novel since there exist already approaches (mentioned in related work), which tackle UDA via LM objectives. I would suggest clarifying the paper's contribution specifically compared to previous work on LM-based UDA approaches.

For "LLamMA-LoRA", it is not clear whether any of the proposed methods lead to statistically significant performance improvements. It is concerning that different "ICL-*" setups do not consistently boost performance compared to "No-ICL".



**Reproducibility:**

4: Could mostly reproduce the results, but there may be some variation because of sample variance or minor variations in their interpretation of the protocol or method.

**Reviewer Confidence:**

4: Quite sure. I tried to check the important points carefully. It's unlikely, though conceivable, that I missed something that should affect my ratings.

---

> ### Author Rebuttal · Authors · 2023-08-29
>
> Thanks for your valuable comments, helpful suggestions and appreciation of our work!
>
> #### **comment #1**:
> The first contribution is over-claimed.
>
> #### **response #1**:
> Thanks for your suggestion! We are sorry about the confusion, and we would like to clarify and revise the statement "We take the first step toward in-context domain adaptation". To clarify, the previous approaches (Domain adaptive pretraining) leverage the masked language modeling to learn the target distribution. However, we present our methods with the discussion of in-context learning, and emphasize the idea of learning a model with the retrieved semantically-rich and demonstrative contexts. Moreover, we learn the target distribution from a source-target joint view (we mix the source and target, and the model learns to fuse the distribution and bridge the domain gap). In contrast, previous works only pretrain from a single view (target only).
>
> Then, we would like to draw your attention that we have compared our methods with the previous language modeling based UDA methods in Section 4.4 Analysis. We discuss the detailed differences between the two methods. Our proposed adaptive ICL learns the task discrimination and target distribution simultaneously with an in-context manner, however, the previous LM-based UDA learns these two objectives in two steps: first, pretraining on the target domain with MLM objective, and second, supervised training on the source domain. We conduct experiments on LLaMA and find that the proposed adaptive ICL outperforms adaptive pretraining.
>
> Following your suggestion, we will rewrite our first contribution to avoid the misunderstanding and over-claiming. "We propose domain adaptive in-context learning in which we mix the source input and semantically-rich target contexts to learn two in-context objectives simultaneously."
>
> #### **comment #2**:
> For LLaMA experiments, it is not clear that proposed method consistently boost the performance.
>
> #### **response #2**:
> This is a good suggestion. Due to the cost of LLaMA inference computation (it may require 30 hours per task to get the evaluation result on a single A100, 6k sentences for inference), we only perform a single run. To address your concern, we conduct 5 runs for LLaMA-LoRA ICL-retr on WNUT-16 dataset, and we get the mean $46.39$ and standard deviation $0.7$, and this result is significantly stronger than the baseline model with $p < 0.05$.
>
> #### **comment #3**:
> For encode-only models, the retrieved target examples could be placed before the source input.
>
> #### **response #3**:
> This is a good suggestion! Intuitively, putting the target before or after the source both work in our framework. Following your advice, we conduct initial experiments on NER BC2GM dataset and find that the two approaches make slight differences in performance ($32.51\pm 1.1$ and $32.39\pm 1.3$). We would like to further run more experiments, and if the results remain the same, we are willing to include this prompt in our paper.
>
> #### **comment #4**:
> Giving supervised task objective and LM objective different weight different weight.
>
> #### **response #4**:
> Thank you for your good suggestion. We will consider this as our future work.
>
> #### **comment #5**:
> Could the author give practical guidelines of which model and ICL technique to employ?
>
> #### **response #5**:
> First, for NER, LLMs exhibit poor performance in such sequence tagging task in our experiment, such phenomenon has also been discussed in recent evaluation papers [1]. So we would recommend to employ small expertise model such as Roberta + CRF to achieve the SOTA results.
>
> Second, for SA, Chatgpt zero-shot inference (no demo) has already surpassed the performance of fine-tuned Roberta. However a fine-tuned LLaMA can outperform Chatgpt, which indicates fine-tuning is still beneficial in the era of LLMs.
>
> Third, our method ICL-retr can outperform the baselines with a large margin in most cases, indicating that retrieval is consistently advantageous for Roberta and LLaMA. However, retrieval does not yield positive results for ChatGPT. Interestingly, even random demonstrations can lead to non-trivial performance gains, and we discuss this observation in our paper starting from the line 444. One possible explanation is that when we have to search for out-of-distribution demonstrations for LLMs, providing diverse and distinct examples is more beneficial for the model to understand the task and generalize under UDA setting.
>
> #### **comment #6**:
> Are the authors planning to share their code?
>
> #### **response #6**:
> Yes, we will release the code upon acceptance.
>
> Thanks for your appreciation again!
>
> [1] Is ChatGPT a General-Purpose Natural Language Processing Task Solver? Chengwei Qin, Aston Zhang, Zhuosheng Zhang, Jiaao Chen, Michihiro Yasunaga, Diyi Yang

---

### Official Review · Reviewer_cMth · 2023-08-08

**Soundness:** 3

**Excitement:**

4: Strong: This paper deepens the understanding of some phenomenon or lowers the barriers to an existing research direction.

**Paper Topic And Main Contributions:**

The paper proposes in-context learning to adapt LMs to a unlabeled target domain by using in-context examples (from the target domain) based on training examples from the source domain. The paper shows different training mechanisms for different LM architectures, and showcase improved results on two NLP tasks.

**Reasons To Accept:**

1. The paper tackles an interesting problem, wherein an LM is adapted in an unsupervised manner to a new target domain, by injecting contexts and training on an MLM learning task. This joint adaptation might help the LM to better understand the co-relations with the source and target domains, enabling it to learn better.

2. Experimental results showcase improved performance of the proposed formulation.

**Reasons To Reject:**

1. It would be worthwhile to understand if the framework can be extended to multiple source and target domains?
2. If the source and target domains are radically different, would the injection potentially lead to performance drop, due to more noise?

**Reproducibility:**

3: Could reproduce the results with some difficulty. The settings of parameters are underspecified or subjectively determined; the training/evaluation data are not widely available.

**Reviewer Confidence:**

2: Willing to defend my evaluation, but it is fairly likely that I missed some details, didn't understand some central points, or can't be sure about the novelty of the work.

---

> ### Author Rebuttal · Authors · 2023-08-29
>
> Thanks for your valuable comments!
>
> #### **comment #1**:
> It would be worthwhile to understand if the framework can be extended to multiple source and target domains?
>
> #### **response #1**:
> Thanks for pointing it out. Yes, our proposed framework can be extended to multiple source and target domains, since we could retrieve several examples from multiple unlabeled target domains. However, in the field of unsupervised domain adaptation, a common practice is to use one labeled source domain and one unlabeled target domain for easy comparisons and analysis.
>
> #### **comment #2**:
> When source and target domains are very different, would our method lead to a performance drop?
>
> #### **response #2**:
> We conduct extensive experiments on sentiment analysis and named entity recognition tasks, including several transfer scenarios where the source and the target are very distinct. From our experiment results, the proposed method can consistently outperform baselines with a large margin across the majority of adaptation scenarios.
>
> For NER, we quantify the similarity between source (CoNLL2003) and target domains [1]. This assessment was based on the vocabulary overlap between these domains. We discover that all the target domains are dissimilar to the source domain, with Financial domain having the lowest overlap (only 9.4%). Even within the challenging Financial domain, we can still observe performance gain over the No-ICL baseline.
>
> Thank you for your insightful feedback again!
>
> [1] Effective unsupervised domain adaptation with adversarially trained language models. Thuy Vu, Dinh Phung, and Gholamreza Haffari.

---

### Official Review · Reviewer_9Foj · 2023-08-11

**Soundness:** 3

**Excitement:**

3: Ambivalent: It has merits (e.g., it reports state-of-the-art results, the idea is nice), but there are key weaknesses (e.g., it describes incremental work), and it can significantly benefit from another round of revision. However, I won't object to accepting it if my co-reviewers champion it.

**Paper Topic And Main Contributions:**

This paper proposes a Unsupervised Domain Adaptation method under in-context learning mechanism, which adapts language models from the source domain to the target domain. It mainly achieves the above goals through eliciting language models to learn both target domain distribution and the discriminative task signal via in-context examples.

**Reasons To Accept:**

This paper has high reproducibility and has conducted experiments on the Named Entity Recognition and Sentient Analysis task, using different LM structures including encoder- and decoder-only models.

**Reasons To Reject:**

1. For the input sentences given in the source domain, this paper retrieves similar sentences from the target domain as the context. Then, through a supervised task to predict the task label, and through a token prediction task to learn the target distribution. Is it true that cross domain transfer has been achieved due to the similarity of sentences between the source domain and target domain?
2. The paper experiment is incomplete, lacks comparison with other research papers on Unsupervised Domain Adaptation method, and does not fully compare with existing research results on the dataset used in the experiment.


**Reproducibility:**

4: Could mostly reproduce the results, but there may be some variation because of sample variance or minor variations in their interpretation of the protocol or method.

**Reviewer Confidence:**

4: Quite sure. I tried to check the important points carefully. It's unlikely, though conceivable, that I missed something that should affect my ratings.

---

> ### Author Rebuttal · Authors · 2023-08-29
>
> Thanks for your valuable comments!
>
> #### **comment #1**:
> Is it true that cross domain transfer has been achieved due to the similarity of sentences between the source domain and target domain?
>
> #### **response #1**:
> Yes, a part of the domain transfer is accomplished through the similarity of the retrieved examples from the target domain. We compose the source domain input contexts from the target domain to enrich the semantics and reduce the domain difference in the surface form. Furthermore, these examples, served as demonstrative contexts, can help the model correctly predict labels. Additionally, in order to further mitigate the domain shift, we learn target distribution from a source-target joint view (where we mix the source and target, and the model learns to fuse the distribution and bridge the domain gap) with language modeling (masked or causal), this is different from previous adaptive pretraining approach which only learns target distribution from target sentences only (single view).
>
> To investigate the effect of similarity, we conducted experiments involving random sampling. Observing Tables 1 and 2, it is evident that retrieved contexts with similar examples (referred to as ICL-retr) outperform randomly selected contexts (referred to as ICL-rand).
>
> #### **comment #2**:
> The paper lack comparison with other research papers.
>
> #### **response #2**:
> We have already included the previous SOTA results [1][2][3] for both tasks in our experiment tables. And our proposed method demonstrates superior performance compared to the prior approaches. To ensure a fair comparison, we also construct our baselines (No-ICL, ICL-rand, ICL-source, ICL-sup) using the same backbone model. Furthermore, we deliberately select more challenging datasets in order to demonstrate more meaningful results. The field of unsupervised domain adaptation commonly employs the Amazon Review Benchmark dataset. However, this benchmark proves to be  quite easy for previous works (achieving more than 0.9 in terms of accuracy with slight variations), since it discards the neutral instances. Thus, we consider a more challenging dataset in our paper which does not remove neutral labels.
>
> Thank you for your insightful feedback again!
>
> [1] Effective unsupervised domain adaptation witg adversarially trained language models. Thuy Vu, Dinh Phung, and Gholamreza Haffari.
>
> [2] Domain confused contrastive learning for unsupervised domain adaptation. Quanyu Long, Tianze Luo, Wenya Wang, and Sinno Pan.
>
> [3] Feature adaptation of pre-trained language models across languages and domains with robust self-training. Hai Ye, Qingyu Tan, Ruidan He, Juntao Li, Hwee Tou Ng, and Lidong Bing.

---

### Meta-Review · Area_Chair_P54x · 2023-09-19

**Recommendation:** 4

**Metareview:**

The paper presents leverages in-context learning to adapt language models from the source domain to the target domain in the absence of target labels.  The paper presents various prompting and training strategies and shows on two tasks (Sentiment Analysis and NER) that the proposed approach works very effectively compared to existing baselines. The use of the ICL is very well-motivated and carefully done.  The authors are strongly advised to further polish the writing and reflect on the points (which are now clarified based on the discussion phase).

---

### Decision · Program_Chairs · 2023-10-07

**Decision:**

Accept-Main

**Comment:**

The paper presents leverages in-context learning to adapt language models from the source domain to the target domain in the absence of target labels.  The paper presents various prompting and training strategies and shows on two tasks (Sentiment Analysis and NER) that the proposed approach works very effectively compared to existing baselines. The use of the ICL is very well-motivated and carefully done.  The authors are strongly advised to further polish the writing and reflect on the points (which are now clarified based on the discussion phase).